# Early Use of ECMO for Refractory Kounis Syndrome Concealed by General Anesthesia—A Case Report

**DOI:** 10.3390/medicina58060759

**Published:** 2022-06-02

**Authors:** Ho Kyung Yu, Miyeong Park, Soo Hee Lee, Jung-Woo Woo, Dong-Hoon Kang, Joung Hun Byun, Seong-Ho Ok

**Affiliations:** 1Department of Anesthesiology and Pain Medicine, Gyeongsang National University Changwon Hospital 11, Samjeongja-ro, Seongsan-gu, Changwon-si 51472, Korea; sciatic@naver.com (H.K.Y.); ele93547@naver.com (M.P.); lishiuji@naver.com (S.H.L.); 2Department of Anesthesiology and Pain Medicine, Gyeongsang National University College of Medicine, Jinju-daero 816 beon-gil, Jinju-si 52727, Korea; 3Institute of Health Sciences, Gyeongsang National University, Jinju-si 52727, Korea; 4Department of Surgery, Gyeongsang National University Changwon Hospital 11, Samjeongja-ro, Seongsan-gu, Changwon-si 51472, Korea; wjfriend11@gmail.com; 5Department of Thoracic and Cardiovascular Surgery, Gyeongsang National University Changwon Hospital 11, Samjeongja-ro, Seongsan-gu, Changwon-si 51472, Korea; drk82@hanmail.net; 6Department of Thoracic and Cardiovascular Surgery, Gyeongsang National University College of Medicine, Gyeongsang National University Changwon Hospital 11, Samjeongja-ro, Seongsan-gu, Changwon-si 51472, Korea; jhunikr@naver.com

**Keywords:** antibiotics, extracorporeal membrane oxygenation (ECMO), Kounis syndrome

## Abstract

A 46-year-old woman demonstrated refractory Kounis syndrome (KS) after induction of anesthesia. Despite conventional management of anaphylaxis and advanced cardiac life support, her cardiovascular function continued to deteriorate until she had a cardiac arrest, and after extracorporeal membrane oxygenation (ECMO) therapy, electrical cardiac activity reappeared. A large number of patients with KS—“allergic angina syndrome”—has been known to recover well with vasodilators; however, this patient showed antibiotics-induced refractory KS during general anesthesia. Severe bronchospasms with desaturation appeared as initial anaphylactic features; however, these did not respond to conventional treatment for anaphylaxis. Patient’s hemodynamic signs eventually worsened, leading to cardiac arrest despite ephedrine administration and chest compressions. During cardiopulmonary cerebral resuscitation, the central line was secured, and epinephrine, atropine, as well as sodium bicarbonate were administered repeatedly; nevertheless, cardiac arrest was sustained. After initiation of veno-arterial ECMO, atrial fibrillation was observed, which was later converted to sinus tachycardia by electrical cardioversions and amiodarone. Coronary angiography was performed before the patient was admitted to the intensive care unit; there were no indications of an impending cardiac arrest. The patient was discharged uneventfully owing to early use of ECMO despite the emergence of KS symptoms that were initially masked by anesthesia but later worsened abruptly.

## 1. Introduction

Patients with an allergic reaction present with variable symptoms ranging from mild skin lesions to severe cardiovascular dysfunction that can lead to death. In this broad spectrum of allergic reactions, anaphylaxis is a life-threatening systemic allergic state [1]. Patients with allergic reactions visiting an outpatient clinic usually present with skin lesions as the initial symptoms. However, anesthetized patients with allergic reactions usually show symptoms indicative of respiratory or circulatory dysfunction as the initial allergic signs because early or mild symptoms of anaphylaxis, such as pruritus, urticaria, dyspnea, or lightheadedness, may be masked by anesthesia or specific conditions related to the surgery. Diagnosis of anaphylaxis based on the clinical criteria can be delayed in the operating room, and it is difficult to make a differential diagnosis with other conditions that present with signs and symptoms similar to those of anaphylactic shock [2].

Many patients with Kounis syndrome (KS), also known as “allergic angina syndrome” [3], have been treated in the absence of specific guidelines in the literature, and an acute coronary syndrome concomitant with an allergic reaction has been reported [4].

We describe the case of a patient who developed a bronchial spasm and critical cardiovascular depression shown by an abnormal electrocardiogram (ECG) after induction of general anesthesia, which was resolved by the early use of extracorporeal membrane oxygenation (ECMO) for refractory KS.

## 2. Case Presentation

A 46-year-old female patient (150 cm, 66 kg) admitted to our hospital was scheduled to undergo hemithyroidectomy under general anesthesia. She had a history of cesarean section 19 years ago and was prescribed amlodipine camsylate, losartan potassium, and nebivolol hydrochloride for hypertension a month before the hemithyroidectomy. During the preanesthetic visit, she denied any history of allergy or acute coronary syndrome. There were no abnormalities in the preoperative ECG, routine laboratory and chest radiologic examination reports, and transthoracic echocardiography (TTE).

The patient was preoxygenated with 100% oxygen, and general anesthesia was induced with 120 mg propofol, continuous remifentanil infusion, and 50 mg rocuronium. After transtracheal intubation, clear breath sounds were checked bilaterally, and mechanical ventilation was initiated. Intravenous cefazolin (1 g) was administered as a prophylactic antibiotic. During skin disinfection and surgical draping, a nurse anesthetist noted high peak inspiratory pressure (PIP, 40 cm H_2_O) with decreased oxygen saturation (78%). A Ventolin inhaler and 5 mg of intravenous dexamethasone were administered assuming that the patient had suffered a bronchial spasm, and manual ventilation was performed with 100% oxygen. A few minutes later, her blood pressure, which was initially at 95/50 mm Hg, dropped to 76/45 mm Hg with 2.5 mV ST segment elevation and severe bradycardia at 36 beats/min. Massive fluid resuscitation and 10 mg of intravenous ephedrine were administered, accompanied by chest compressions for cardiopulmonary cerebral resuscitation (CPCR); however, her blood pressure continued to decrease to an undetectable level on noninvasive monitoring. Despite two intravenous injections of 1 mg epinephrine, administration of 0.5 mg atropine at three-minute intervals, as well as continuous CPCR, her heart rate decreased further, and the ECG showed sinus arrest. During CPCR, an ultrasonography-guided internal jugular vein was secured for continuous administration of nitroglycerine and epinephrine. Forty minutes after sinus arrest, veno-arterial extracorporeal membrane oxygenation (VA-ECMO) was initiated upon emergency consultation with cardiac surgeons. Arterial blood gas analysis of a sample of blood taken from the femoral artery before ECMO initiation showed a pH of 7.25, PaCO_2_ of 47 mm Hg, PaO_2_ of 84 mm Hg, and oxygen saturation of 87%. Global akinesia was found on transesophageal echocardiography. Ten minutes after initiation of VA-ECMO, ECG findings revealed that the sinus arrest had converted into atrial fibrillation. Sinus rhythm (145 beats/min) was re-covered after three applications of 200 J defibrillation in biphasic mode followed by CPCR, and the PIP decreased to 20 mm Hg. During electrical conversion, 150 mg of amiodarone were slowly administered intravenously, and a continuous infusion of amiodarone was initiated. The left radial artery was secured for continuous hemodynamic monitoring and laboratory examination. At this time, the plasma glucose, sodium, and potassium levels were 287 mg/dL, 145 mmol/L, and 3.4 mmol/L, respectively. During CPCR and VA-ECMO application in the operating room, 260 mEq sodium bicarbonate, 4.5 mg epinephrine, and 2.5 mg atropine were infused intermittently (Figure 1).

The scheduled hemithyroidectomy was postponed, and the patient was admitted to the intensive care unit (ICU) while still on VA-ECMO. Coronary angiography (CAG) was performed before admission to the ICU, and there was no significant coronary stenosis that could have led to acute coronary syndrome (Appendix A). The patient’s serum creatine kinase-MB (CK-MB) and troponin-T levels increased to 33 ng/mL and 722 ng/mL, respectively. Her husband mentioned a history of analgesic-related allergy; however, he did not know the exact name of the culprit drug. On the second day of ICU admission, ECG showed an ST-T abnormality indicative of anterior ischemia. However, bedside TTE showed normal left ventricular function without any abnormality in the regional wall movement. VA-ECMO was switched to veno-venous ECMO, and the patient was weaned off ECMO on the fifth day of ICU admission. A colleague at the hospital where the patient worked informed us that she had previously experienced severe anaphylaxis with either a cephalosporin antibiotic or ketorolac. The patient was weaned off mechanical ventilation a day after ECMO weaning, and the endotracheal tube was removed. She was transferred to the general ward on the seventh day of ICU admission and was discharged on the 12th day of admission to the general ward; her ECG at discharge was normal.

Forty-three days after discharge, she was readmitted for delayed hemithyroidectomy. Preoperative skin prick tests for anesthetics, neuromuscular blockers, opioids, and acetaminophen, which were planned to be used during the perioperative period, were negative. She underwent hemithyroidectomy without any complications.

## 3. Discussion

The cumulative incidence of anaphylaxis in several studies with different follow-up periods is reported to be 26–54%, with the majority reporting the incidence of less than 35%. In them, the incidence of fatal anaphylaxis was 0.21 to 1.06 per million per year, with the mortality rate of 0.3% to 2%, which is very low [5]. General physicians, and especially anesthesiologists, tend to be extremely cautious while prescribing drugs to patients with a history of allergy to certain medications. It is because during anesthesia, early or mild symptoms of anaphylaxis, pruritus, urticaria, dyspnea, or lightheadedness are easily masked. Therefore, patients with a history of allergic reactions need to be monitored with extra care by anesthesiologists in the operating room. Common allergens during the perioperative period include neuromuscular blockers, antibiotics, latex, dyes, colloids, antiseptics, iodinated contrast media, and aprotinin [6]. The overall incidence of anaphylaxis is about 0.01–0.02% during anesthesia; however, 3% of anesthesia-related deaths involved anaphylaxis in France, and 10% of perioperative-period hypersensitivity reactions were fatal in the United Kingdom [7]. Most drugs used in the perioperative period can cause anaphylaxis, most of which are administered intravenously and might cause more serious allergic reactions.

In the US, the most common cause of fatal anaphylaxis was drugs, followed by unknown substances, venom, and food between 1999 and 2010. Only 25% of the agents responsible for fatal anaphylaxis caused by drugs were identified, and 40% of the drugs, which could be identified, were antibiotics [8]. However, the main causative agent for drug-induced fatal anaphylaxis reported in the UK was a general anesthetic [9]. These general anesthetics and antibiotics are administered to patients during induction of anesthesia and often cause allergic reactions. Anaphylaxis is the most critical systemic allergic reaction, and the criteria for diagnosing anaphylaxis suggested by the World Allergy Organization are the presence of any of the following two conditions: (i) acute urticaria, pruritus, or flushing on the skin, mucosa, or both regions, with respiratory abnormality or decreased blood pressure-associated syncope or hypotonia, or severe gastrointestinal symptoms (cramps or vomiting); (ii) acute hypotension, bronchospasm, or laryngeal involvement after exposure to a known or highly suspected causative agent, without skin lesions [1]. Applying these criteria to anesthetized patients is difficult, especially under general anesthesia. Effective management of our patient was even more difficult as the history of previous drug allergy was not shared during the pre-anesthesia visit.

Kounis and Zavras suggested that “allergic angina syndrome (ACS)” as a coronary spasm or acute myocardial infarction (AMI) are induced by an allergic reaction, and the syndrome was named “KS” [3]. During general anesthesia, early and mild symptoms of allergic reactions, such as pruritus, urticaria, dyspnea, or lightheadedness may be masked by anesthesia and the surgical drape, but close patient monitoring through visual inspection, hemodynamic monitoring, and evaluation of respiratory function parameters may help identify allergic features of KS. Careful preanesthetic history-taking, including history of medication use and allergies, especially is essential in suspecting KS. If KS is suspected owing to the emergence of allergic features with acute coronary syndrome, such as worsened hemodynamic monitoring parameters with an ST segment change in the electrocardiogram (ECG), additional laboratory, echocardiographic, and coronary angiographic examination are needed for differential diagnosis and confirmation of KS [10]. There are three types of KS: coronary artery spasm with or without an increase in cardiac enzymes, coronary artery spasm with plaque erosion or rupture leading to AMI, and coronary artery stent thrombosis due to an allergic reaction. Type 1 variant is the most common, seen in approximately 72% of the cases, followed by type 2 in 22%. Generally, most patients with KS recover completely as type 1 is the most common, and the vasospasm seen in this type can be easily reversed with vasodilators [10]. Recently, there have been some reports regarding the use of ECMO in refractory Kounis syndrome (KS) [11,12,13]. These cases occurred in a setting where coronary angiography could be performed. However, we believe that it is very difficult to diagnose KS while the patient is under general anesthesia and to contact the ECMO team during the early period of advanced cardiovascular life support. In our patient, type 1 KS with normal CAG was refractory to conventional anaphylaxis treatment with nitroglycerine. The first-line treatment for anaphylaxis is intramuscular epinephrine, and many physicians follow this protocol. In KS, epinephrine and beta blockers should be used with caution because the former can worsen MI, prolong the QTc interval, and induce a coronary spasm while the latter can aggravate the coronary spasm by increased activity of alpha receptors [10]. For patients on beta blocker therapy preoperatively, as in our patient, epinephrine administered for anaphylaxis can cause a more severe coronary spasm due to the relatively increased activity of alpha receptors, especially when infused intravenously [4]. Initially, our patient showed a bronchial spasm indicative of an allergic reaction and was treated with a bronchodilator, an intravenous steroid, and a high concentration of oxygen; however, her vital signs worsened to hypotension and severe bradycardia in a few minutes. If her past history of anaphylactic reactions to cephalosporine and NSAIDs had been known and prophylactic antibiotics had been administered, we would have infused a lower dose of epinephrine intravenously during CPCR. However, the CPCR performed after sinus arrest and before VA-ECMO seemed to maintain adequate ventilation and perfusion. VA-ECMO appeared to have maintained coronary circulation more effectively than manual chest compressions [14]; furthermore, intravenous heparin administered for ECMO may have relieved the coronary artery thrombus that might develop during KS. The sinus arrest on ECG transformed into atrial fibrillation, and the patient regained sinus rhythm following three attempts at defibrillation. Ridella and Bagdure asserted that the treatment of antibiotic-induced KS should include control of anaphylaxis and vasospasmolytics, and epinephrine should be used with caution [15]. Hence, early use of ECMO should be considered in the management of antibiotic-induced refractory KS.

The causative drugs for KS include antibiotics, antivirals, antifungals, NSAIDs, proton pump inhibitors, antihistamines, anesthetics, anti-neoplastic drugs, neuromuscular blockers, and others [10]. Causative drugs used in our patient prior to the onset of symptoms were propofol, rocuronium, and cefazolin. We did not suspect cefazolin as the cause of refractory KS in the operating room. However, subsequently, the detailed history of anaphylaxis provided by the patient’s colleague regarding repeated skin rash after antibiotic administration during ICU stay (antibiotics skin test was negative) led us to suspect cefazolin as the cause of anaphylaxis.

A previous report from Europe indicated that cephalosporin, which was used in our patient as a prophylactic antibiotic, has a relatively lower risk of anaphylaxis than dextran, contrast media, and blood, which may be used during the perioperative period [16]. In the US, cases of drug-induced fatal anaphylaxis increased twofold in the last decade, and the proportion of antibiotics as the cause of fatal anaphylaxis has gradually increased, while the proportion of unspecified drugs causing anaphylaxis has decreased [8]. Allergic reactions caused by prophylactic antibiotics have increased ninefold since the first guidelines for preventive antibiotic use were published in 1994 [6]. Hence, anesthesiologists should be aware of fatal anaphylaxis that might be caused by disinfections, latex, and antibiotics, in addition to anesthesia-related drugs and should be cautious when monitoring anesthetized patients.

The following points should be borne in mind during the administration of anesthesia in patients at risk of drug-induced anaphylaxis.

Anesthesiologists should remember that various anesthetics, antibiotics, and other drugs administered together during anesthesia induction can induce severe anaphylaxis and cause KS (ACS can cause fatal acute coronary syndrome).KS should be managed in both allergic reactions and acute coronary syndrome. According to the World Allergy Organization guidelines, intramuscular epinephrine (0.01 mg/kg of body weight, to the maximum total dose of 0.5 mg by 1 mg/mL (1:1000) epinephrine) is the first-line drug for anaphylaxis treatment [1]. Epinephrine as the first drug of choice in anaphylaxis should be cautiously administered with close hemodynamic monitoring, especially when injected intravenously. Intravenous epinephrine should be cautiously used at a dilution of 1:10,000 to 1:100,000 [17].Early initiation of ECMO in refractory KS with conventional treatment and CPCR should be considered because ECMO can maintain coronary circulation more effectively [14], and intravenous heparin with ECMO may be helpful owing to its anticoagulation effect during myocardial infarction.

## 4. Conclusions

Anesthesiologists should keep in mind that various drugs administered in the perioperative period can cause KS, which can be masked by general anesthesia. Epinephrine, the first choice of treatment for anaphylaxis, should be used cautiously as it can worsen coronary symptoms. Above all, the use of ECMO should be considered early on in the management of patients with antibiotic-induced severe KS.

## Figures and Tables

**Figure 1 medicina-58-00759-f001:**
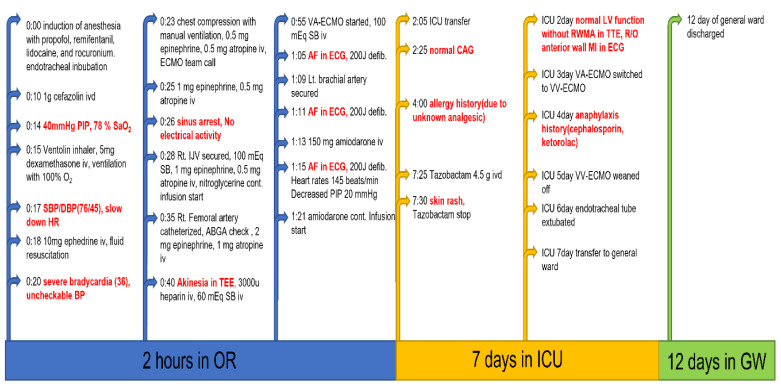
Timeline of refractory Kounis syndrome progression and recovery. OR: operating room, ICU: intensive care unit, GW; general ward.

## Data Availability

The study did not report any data.

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
