# Peer review of "Early Use of ECMO for Refractory Kounis Syndrome Concealed by General Anesthesia—A Case Report"

_medicina, 2022, doi:10.3390/medicina58060759_

Round 1
Reviewer 1 Report
Manuscript Number: medicina-1712850
Title: Early use of ECMO for refractory Kounis syndrome – case report
Author: Hokyung Yu MD, etc.
- General Comments
This manuscript, the author investigated a valuable case of ECMO for Kounis syndrome. Although the findings are of interest, their appeal is too limited warrant publication.
- Major Comments
- P8L196-199
If there is no evidence for the dose of epinephrine, is it not that an overstatement? If there is evidence, please provide reference.
2)
It is not clear from the title and conclusion whether the paper is about early introduction of ECMO for KS or about the need for caution in KS during anesthesia. Please reconsider the title and conclusion.
Author Response
- Major Comments
- P8L196-199
If there is no evidence for the dose of epinephrine, is it not that an overstatement? If there is evidence, please provide reference.
We thank the reviewer for the meticulous review of our paper and for the beneficial suggestion.
KS should be managed in first in an allergic reaction. Intramuscular epinephrine 0.2-0.5mg (1:1000) is the treatment of choice for allergic reaction, however, if this dose is administered intravenously, it can aggravate myocardial ischemia, ventricular arrhythmia, and induce a hypertensive crisis (all of which are adverse effects of epinephrine). While managing anaphylaxis, many physicians are often confused regarding the epinephrine dosage or dilution rate. We have rephrased this point as follows (Line 239 of the revised manuscript):
“According to the World Allergy Organization guidelines, intramuscular epinephrine (0.01 mg/kg of body weight, to a maximum total dose of 0.5 mg by 1 mg/mL (1:1,000) epinephrine) is first-line drug for anaphylaxis treatment.”
“Intravenous epinephrine should be used at a dilution of 1:10,000 to 1:100,000 cautiously.” [1,2]
2)
It is not clear from the title and conclusion whether the paper is about early introduction of ECMO for KS or about the need for caution in KS during anesthesia. Please reconsider the title and conclusion.
We wanted the convey both these points. Anesthesia can mask the typical chest pain of acute coronary syndrome induced by KS and the conventional drug management (1mg epinephrine every 3-5 min) in ACLS (advanced cardiovascular life support) could aggravate myocardial ischemia even if the patient is in asystole. In the Discussion, we have described the difficulties in diagnosis and management of KS during anesthesia and have stressed the need for early consideration of ECMO in refractory KS. Accordingly, we have corrected the title as follows:
“Early use of ECMO for refractory Kounis syndrome concealed by general anesthesia – A case report”
We have additionally revised the Conclusion to convey both these points.
Reviewer 2 Report
Yu et al. describe an interesting case of refractory Kounis-Syndrome leading to the need of V-A ECMO implantation. Although Kounis-Syndrome has been well described, with several published manuscripts on this subject, unique in this case is the need of V-A ECMO use. The discussion should provide more information about the use of V-A ECMO in cases of refractory Kounis-Syndrome, based on literature research. Other reports of V-A ECMO use for Kounis-Syndrome already published? The reported percentages regarding anaphylaxis and mortality rates in the discussion section, should be more clearly presented (reference to which operation category? overall/all operations, thyroidectomy? etc). The manuscript needs thorough language editing to improve text clarity and comprehensibility (e.g. lines 21-23 and abstract section overall, lines 124-125).
Author Response
We thank the reviewer for the thorough review of our manuscript and for the valid comments.
Recently, there have been some reports about the use of ECMO in refractory Kounis-Syndrome (KS)[3-5]. These cases occurred in a setting where coronary angiogram could be performed. In our humble opinion, we believe that it is very difficult to diagnose KS while the patient is under general anesthesia and to contact ECMO team during the early period of advanced cardiovascular life support. We have incorporated this information the Discussion section as per the recommendations (Line 186–191 of the revised manuscript).
Furthermore, the standard dose of intravenous epinephrine for cardiopulmonary cerebral resuscitation (CPCR) could aggravate myocardial ischemia in KS, therefore, reports of similar clinical cases occurring in the operation room (OR) would have been disastrous, and this could possibly be the reason why a lot of similar cases have not been reported. Therefore, this case is a rare clinical report regarding the use of ECMO for KS in an OR setting.
We have revised previous misrepresentations about the incidence of anesthesia-related anaphylaxis and the rate of mortality. The overall incidence and mortality, originally reported, were anesthesia-related. Therefore, in our humble opinion, we believe that the relationship of KS with the type of surgery performed would not have been relevant.
Lines 147–150 of the revised manuscript: “The overall incidence of anaphylaxis is about 0.01-0.02% during anesthesia; however, 3% of anesthesia-related death involved anaphylaxis in France, and 10% of the perioperative-period hypersensitivity reactions were fatal in United Kingdom [7].”
We have revised lines 21-23. We agree that there were several language and grammar-related errors in the Abstract and in some places in the main text. The manuscript was submitted to a professional English editing service (www.editage.co.kr). All language and grammar related errors have been corrected in the revised manuscript. We have ensured logical coherency and improved flow of ideas.
Lines 140–142 of the revised manuscript (originally lines 124–125): “General physicians, and especially anesthesiologists, tend to be extremely cautious while prescribing drugs to patients with a history of allergy to certain medications."
Reviewer 3 Report
The authors report an interesting case of an intraproceduarl anaphylactic reaction presumably due to the administration of cefazoline complicated by acute cardiopulmonary failure. The patient was treated with V-A ECMO.
The reviewer agrees with the authors that general anesthesia masks a multitude of allergic symptoms hindering the diagnosis in the operation theater and that the Kounis syndrome is of utmost interest to the providers during surgery.
However after reading the manuscript remarks need to be made:
- the abstract and the introduction needs to be revised. The general symptoms of anaphylaxis are well known. The authors should provide a general overview of the Kounis syndrome and the difficulties of diagnosing KS under general anesthesia and not a generic description of symptoms of allergic reaction.
- How exactly was KS diagnosed in this case? A picture of the coronary angiogram would be relevant.
- In the discussion, the authors should provide a general diagnostic work flow for the Kounis syndrome under general anesthesia
- The manuscript needs rigorous editing and revision of English language and style.
- There are numerous typos (i.e. line 20, line 49, line 50, line 52, line 103).
- Line 124 - 127: please revise, not clear
- line 133 - 138 are not relevant for this case in my opinion
- line 198 - 199 please provide the correct dose of epinephrine
- line 200 - 202 please provide a reference for this statement
Author Response
However after reading the manuscript remarks need to be made:
- the abstract and the introduction needs to be revised. The general symptoms of anaphylaxis are well known. The authors should provide a general overview of the Kounis syndrome and the difficulties of diagnosing KS under general anesthesia and not a generic description of symptoms of allergic reaction.
Response:
We thank the reviewer for the kind evaluation of our manuscript and for the insightful comments. The abstract has been considerably revised to ensure logical coherency and correct use of language and grammar. We have incorporated additional description regarding KS with respect to diagnosis and difficulties of diagnosis of KS during general anesthesia in the Discussion section highlighted in green color.
- How exactly was KS diagnosed in this case? A picture of the coronary angiogram would be relevant.
Response:
We have attached our patient’s coronary angiogram as a supplemental figure, and there was no significant stenosis or occlusion in the coronary artery.
- In the discussion, the authors should provide a general diagnostic workflow for the Kounis syndrome under general anesthesia.
Response:
We added in brief the diagnostic workflow for the Kounis syndrome under general anesthesia in the Discussion on lines 171–181 of the revised manuscript.
- The manuscript needs rigorous editing and revision of English language and style.
Response:
Our revised manuscript was edited by a professional English editing service, Editage (www.editage.co.kr) to correct the language and grammar to improve comprehensibility of the paper. We agree that there were several language and grammar-related errors in the Abstract and in some places in the main text. We have ensured logical coherency and improved flow of ideas.
We have herewith attached the editing certificate.
- There are numerous typos (i.e. line 20, line 49, line 50, line 52, line 103).
Response:
We have corrected these typographical errors and highlighted the changes in green color.
- Line 124 - 127: please revise, not clear
Response:
We revised the sentence as follows (Lines 140–145 of the revised manuscript):
“General physicians, and especially anesthesiologists, tend to be extremely cautious while prescribing drugs to patients with a history of allergy to certain medications. It is because during anesthesia, early or mild symptoms of anaphylaxis, pruritus, urticaria, dyspnea, or lightheadedness are easily masked. Therefore, patients with a history of allergic reactions need to be monitored with extra care by the anesthesiologists in the operating room.”
- line 133 - 138 are not relevant for this case in my opinion
Response:
We have deleted first sentence (line 133-138 in the original manuscript) that the reviewer has pointed out. The next sentence presented that the major cause of fatal anaphylaxis was drugs and among them, antibiotics accounted for the largest proportion of drugs in US. Additionally, these antibiotics are being increasingly infused in patients prior to anesthesia. We thought, in our humble opinion, which these points need to be highlighted so that anesthesiologists realize the need to pay close attention to administration of antibiotics and anesthetics and consider them as causative drugs for KS during anesthesia.
- line 198 - 199 please provide the correct dose of epinephrine
Response:
According to the World Allergy Organization guidelines, intramuscular epinephrine (0.01 mg/kg of body weight, up to a maximum total dose of 0.5 mg made by 1 mg/mL (1:1,000) epinephrine) is first-line drug for anaphylaxis treatment. In conventional advanced cardiovascular life support, 1 mg epinephrine every 3-5 min is recommended through intravenous or intraosseous route repeatedly. If intravenous epinephrine is infused with high concentration (1:1,000), it will be very difficult to control the precise dose and myocardial ischemia could be aggravated easily. Therefore, in patients with severe Kounis-Syndrome who need CPCR, epinephrine should be administered intravenously in a dilution of 1:10,000 to 1:100,000, and it may be repeated depending on the patient's condition.
We described the importance of hemodynamic close monitoring, and fine control of epinephrine dosage using more diluted epinephrine (Lines 239–242 of the revised manuscript).
- line 200 - 202 please provide a reference for this statement
Response:
ECMO maintained a higher coronary perfusion pressure in a refractory ischemic cardiac arrest animal model than in conventional resuscitation [6]. We have incorporated this reference in the Discussion (Lines 245 of the revised manuscript).
We have also revised an incorrect expression regarding the anticoagulant property of heparin, and have emphasized that it could contribute to aggravation of Kounis-Syndrome presenting with myocardial infarction (Lines 245–246 of the revised manuscript).
Reference
- Stub, D.; Byrne, M.; Pellegrino, V.; Kaye, D.M. Extracorporeal Membrane Oxygenation to Support Cardiopulmonary Resuscitation in a Sheep Model of Refractory Ischaemic Cardiac Arrest. Heart, Lung and Circulation 2013, 22, 421-427, doi:https://doi.org/10.1016/j.hlc.2012.11.020.
Round 2
Reviewer 1 Report
Manuscript Number: medicina-1712850
Title: Early use of ECMO for refractory Kounis syndrome – case report
Author: Hokyung Yu MD, etc.
Thank you for your kind correction. I have no additional correction.
Reviewer 3 Report
I have no further comments